# Enhancing generalizability of deep networks via Fisher regularization

## Abstract

The generalization ability of a deep learning classifier hinges significantly on the geometry of its loss landscape. Solutions residing near flatter areas are more robust, generalizing better than the ones present near sharp minima. In this paper, we study the effects of the loss landscape on the generalization of deep learning models and effectively leverage its geometric information to propose a novel regularization method, Fisher regularization. By dynamically penalizing weights based on their curvature across the loss landscape, we propose an *adaptive* regularization scheme that guides the optimization process towards flatter and more generalizable solutions. We establish a rigorous theoretical foundation for our regularization approach using the PAC-Bayesian theory and empirically validate the superior performance of deep learning models trained with our proposed method over other powerful regularization techniques across a range of challenging image classification benchmarks.

## 1 Introduction

Building machine learning models that perform well on unseen data is crucial to ensure that the model generalizes well to different variations of the data distribution while avoiding overfitting the data it is trained on. Although SGD (Stochastic Gradient Descent), a well-known method for training deep neural networks, can effectively navigate the loss landscape to reach global minima, it is well known that not all minima are created equal (Liu et al., 2021). Their generalization properties also depend on other properties such as their *flatness* (Hochreiter & Schmidhuber, 1997) and volume. Despite their remarkable success in various applications, a deeper theoretical understanding of the *generalizability* of deep neural networks based on the properties of the loss landscape and the minima reached by SGD is still lacking.

Although the theoretical study of generalization provides valuable insight, it is equally important to explore practical methods that effectively improve the generalizability of deep networks. *Regularization* is a popular framework for improving the generalization of deep networks. Regularization strategies based on the weight norm, such as the $l_2$ norm and the $l_1$ norm or lasso (Tibshirani, 1996), are commonly employed in deep networks. However, these techniques typically operate on the weights themselves and do not take into account the flatness or curvature of the loss landscape. They penalize all weights equally, without taking into account their contributions to the *sharpness* of the loss landscape (Hochreiter & Schmidhuber, 1997), which are key factors in determining generalization.

In this work, we analyze generalizability from the viewpoint of the loss landscape, exploring how its geometric properties can be exploited to improve regularization techniques. To this end, we propose Fisher regularization, a novel regularization method that considers the geometry of the loss landscape to determine the relative importance of different weights in the regularization term.

To provide a theoretical grounding for Fisher regularization, we leverage the PAC-Bayesian framework (Dziugaite & Roy, 2017) to derive a novel generalization bound. The bound is based on the Fisher-norm, a well-established measure of model sharpness. This analysis not only motivates our regularization method, but also connects it to the perspective that considers *sharpness* as an effective measure of generalization.

We confirm the effectiveness of our Fisher regularization through extensive experiments on several challenging image classification datasets. Our results demonstrate improved accuracy and generalization of models trained with Fisher regularization compared to state-of-the-art regularization methods.

To explain our approach and to demonstrate its effectiveness in improving generalization, we have organized the paper as follows. In Section 2, we present related works that have explored different methods in deep learning to improve the generalization of neural networks. Section 3 provides theoretical motivation for our approach by connecting it to a generalization bound based on the PAC-Bayesian framework and demonstrates how the Fisher norm minimizes the bound to improve its performance. Establishing this connection provides a theoretical grounding for the effectiveness of our approach. Section 4 then describes our regularization method in detail, and Section 5 provides details about the experiments. Section 6 then compares Fisher regularization with other regularization techniques, and finally, section 7 concludes with potential future directions.

Overall, our work makes the following contributions.

- We propose a novel regularization method, called Fisher regularization, which incorporates the sharpness of the loss landscape into the regularization term to improve the generalization of neural networks. This is significant because, unlike $L_2$ regularization, which applies uniform penalties to all weights, our method penalizes the weights in proportion to their curvature on the loss landscape. This adaptive weighting scheme allows our method to focus more on penalizing parameters that contribute to the curvature of the loss landscape, and hence are more prone to overfitting.

- We successfully leverage the PAC-bayesian framework, which provides non-vacuous and tight generalization bounds, to derive our Fisher regularization term. We thus provide a robust theoretical foundation for our approach based on the learning-theoretic framework of the generalization gap.

## 2 Related Works

**Generalization bounds in deep learning**
An important quantity that estimates the generalization ability of a classifier is its generalization bound. These bounds depend on the difference between the empirical risk of a classifier on a dataset and its true risk and measure how well the classifier, when trained on a training set, will generalize to an unseen test set. In the literature, many different generalization bounds have been suggested for neural networks using the uniform convergence principle, which are based on the fat-shattering dimension (Bartlett, 1998), covering numbers, or the VC dimension (Bartlett & Maass, 2003). As a consequence of these, many methods have been introduced that minimize these bounds to improve generalization of deep networks, such as $L_2$ norm regularization that minimizes the fat-shattering dimension (Bartlett, 1996), and $L_0$ norm regularization which minimizes the VC dimension (Bartlett & Maass, 2003) by minimizing the number of hidden units. However, recent works (Nagarajan & Kolter, 2019) have shown that, due to their dependence on the uniform convergence principle, these bounds are essentially vacuous and thus do not explain generalization in neural networks.

On the other hand, a line of recent works have explored PAC-Bayesian bounds to provide generalization bounds that are non-vacuous (Dziugaite & Roy, 2017; Neyshabur et al., 2018). These bounds usually contain terms based on the properties of the loss landscape, such as *sharpness*. In this work, we thus use the concepts from the PAC-Bayesian theorem with a more sound theoretical basis to suggest a novel regularization method that considers the geometry of the loss landscape into account.

**Regularization in deep learning**
Regularization is a common technique that is widely used to enhance the generalization capabilities of deep neural networks. These methods usually employ some form of penalty on the complexity of neural networks to mitigate their risk of overfitting. A significant portion of these methods apply a penalty on the norm of the network weights to prevent them from growing too large. Weight decay (Zhang et al., 2018), early stopping (Bai et al., 2021), and $l_p$-norm penalties are some popular strategies that fall into this category. The $l_p$

norm regularization is the most popular method that is widely used in different deep learning architectures to improve their generalization performance.

Several variants of these methods exist that apply different forms of penalties on the weights, serving different purposes. For example, $L_2$ regularization is widely used to minimize the $L_2$ norm of the weights, whereas $L_1$ regularization (popularly known as the Lasso (Tibshirani, 1996)) is employed to encourage sparsity of the weights, driving less influential weights to zero and thus performing feature selection. Beyond individual weight penalties, $L_{2,1}$ regularization extends this concept to group-level sparsity (Scardapane et al., 2017). It uses a regularization term that takes the $L_2$ norm of the columns of the weight matrices and then sums these norms, essentially computing an $L_1$ norm of the column norms. Since the weights in each column include the outgoing weights from the same hidden unit, this regularization method effectively prunes entire hidden units by zeroing out all the terms in the corresponding column, thus providing a form of structure regularization. On the other hand, *spectral* norm regularization (Yoshida & Miyato, 2017) reduces the sensitivity of the network output to small perturbations by penalizing the spectral norm, which corresponds to the largest eigenvalue of the weights. This constraint limits the Lipschitz constraint of the network, thus improving its adversarial robustness and generalization capability.

## 3 An improved data-dependent PAC-Bound

The generalization gap of a predictor $f$ can be described as the difference in its empirical risk $R_n(f) = \frac{1}{n}\sum_{i=1}^{n} L(x,y)$ over the training set and its expected risk (or true risk) $R(f) = \mathbb{E}[L(x,y)]$. This gap, which can be written as $R(f) - R_n(f)$, is a predictor of how well a network that is trained on the training set, will generalize to unseen test data sampled from the same distribution $P(X, Y)$. If the input dataset contains samples $X_1, X_2, ..., X_n$ with labels $y_1, y_2, ..., y_n$ and if our risk function is the margin loss (Bartlett et al., 2017) which is described as

$$L(x,y) = \begin{cases} 0 & y * f(x) \geq \gamma \\ 1 - \frac{y*f(x)}{\gamma} & y * f(x) \in [0, \gamma] \\ 1 & y * f(x) < 0 \end{cases} \tag{1}$$

where $\gamma$ is the margin between the classes. Using the PAC-Bayesian generalization error bound (Dziugaite & Roy, 2017), given a prior $P$ over the weights $W$ and letting the posterior distribution of the weights output by a learning algorithm $\mathcal{A}$ be $Q(S)$, then with probability $1 - \delta$ this gap can then be bounded as

$$R(f) - R_n(f) < 4\sqrt{\frac{(KL(Q\|P) + ln(m/\delta)}{m-1}} \tag{2}$$

In the regular PAC-Bayesian bounds, the prior $P$ is usually a Gaussian distribution with mean fixed at 0 and covariance $\mathbf{I}$.

If the weight distribution is a Gaussian with mean $w$ and variance $\Sigma_q$, and the prior is a normal distribution centered at $u$ with covariance $\Sigma_p$, the above bound can be rewritten as

$$R(f) - R_n(f) < 4\sqrt{\frac{(KL(\mathcal{N}(w, \Sigma_q)\|\mathcal{N}(u, \Sigma_p)) + ln(m/\delta)}{m-1}} \tag{3}$$

where $m$ is the number of samples in the test set.

Using the following Equation from (Martens, 2020)

$$KL(P(w)\|P(u)) = (w - u)F(w - u)^T \tag{4}$$

where $F = \mathbb{E}[(\partial \mathbf{L}/\partial \mathbf{w})(\partial \mathbf{L}/\partial \mathbf{w})^{\mathbf{T}}]$ is the fisher information matrix, we can derive a tighter generalization bound that is based on the geometric information of loss landscape, captured via this matrix. Since the weights in most neural networks are initialized from a zero mean normal distribution, we put $u = 0$ in the above equation, and the bound in (3) simplifies to

$$R(f) - R_n(f) < 4\sqrt{\frac{\|\mathbf{w}\mathbf{F}\mathbf{w}^{\mathbf{T}}\| + ln(m/\delta)}{m-1}} \tag{5}$$

Thus, the value contributed by the KL term to the bound can be broken down into two parts: a) the magnitude of the weights $\mathbf{w}$, which is multiplied by b) their expected Fisher information $\mathbf{F}$. The Fisher information can be understood as the sharpness of the loss landscape corresponding to each weight, and determines how much the individual weights contribute to the complexity of the network.

This motivates a novel method to regularize deep networks by minimizing the norm of the product of weights and their Fisher information. We thus design deep convolutional support vector machine (SVM) models (Tang, 2013) that use this data-dependent regularization method to penalize the weights and achieve improved generalization. We now describe our method in further detail.

## 4 Deep Fisher regularized large margin networks

Inspired by the generalization bound stated above, we formulate a variant of the deep support vector machine with a novel regularizer term that minimizes the norm of the product of the weights with the Fisher information to learn a smooth decision hyperplane that separates points in different classes by a margin $\gamma$. More formally, denoting the margin loss defined in (1) as $L_\gamma$, our SVM model minimizes the following loss function:

$$\sum_{i=1}^{n} L_\gamma(x_i, y_i) + \beta \sum_{i=1}^{n} \|\mathbf{w}\mathbf{F}\mathbf{w^T}\| \tag{6}$$

where $\mathbf{F}$ is the Fisher information matrix, $L_\gamma(x_i, y_i)$ denote the margin loss over the data, $\beta$ is the weightage given to the Fisher regularization. Replacing the Fisher information with its average over the mini-batch, this loss can be finally written as:

$$\sum_{i=1}^{n} L_\gamma(x_i, y_i) + \beta \sum_{i,j=1}^{n} \|w_i \cdot (\sum_{m=1}^{M} \frac{\partial L(x_m, y_m)}{\partial w_i} \cdot \frac{\partial L(x_m, y_m)}{\partial w_j}) \cdot w_j\|^2 \tag{7}$$

where $w_i$, $w_j$ denote the $i_{th}$ and $j_{th}$ elements of the weight vector $\mathbf{w}$, $\frac{\partial L}{\partial w_i}, \frac{\partial L}{\partial w_j}$ their gradients, and the sum of their products over a batch of size $M$ gives us the Fisher information matrix. In this equation, the second term forces the loss landscape to be flat by minimizing the magnitude of each weight by a factor proportional to *curvature* of the loss landscape under its variation. This will thus tend to encourage the model to converge to a minima where the loss will be robust to small perturbations of the weights. Moreover, our proposed regularization method encourages the generalization gap to be small by directly minimizing the PAC-Bayesian bound described in Equation (5).

Thus, our method gives a higher *importance* to the weights in the regularization penalty that correspond to directions of higher curvature on the loss landscape. This makes the regularization scheme adaptive and more grounded in theoretical foundations, while also guiding the optimization towards minima that are *flatter* with better generalization properties. We use this novel regularization term to train deep convolutional SVM (support vector machines) models and demonstrate an improved generalization achieved by this method over other regularization schemes.

## 5 Experiments

### 5.1 Datasets

We tested our models on the MNIST digit (LeCun et al., 2004) and the CIFAR-10/100 datasets (Krizhevsky, 2009). The MNIST dataset consists of grayscale images of 10 handwritten digits and consists of 60000 images of size $28 \times 28$ for training and 10000 images as test examples. It is divided into 6 training batches and 1 test batch, each containing 10000 images (Simonyan & Zisserman, 2014). We performed all our experiments on the training set and used the test set to test the performance and generalization of the trained models.

The CIFAR-10 dataset (Simonyan & Zisserman, 2014) consists of 60000 $32 \times 32$ color images belonging to 10 categories with 6000 images per category. It is divided into 5 training batches and 1 test batch, each

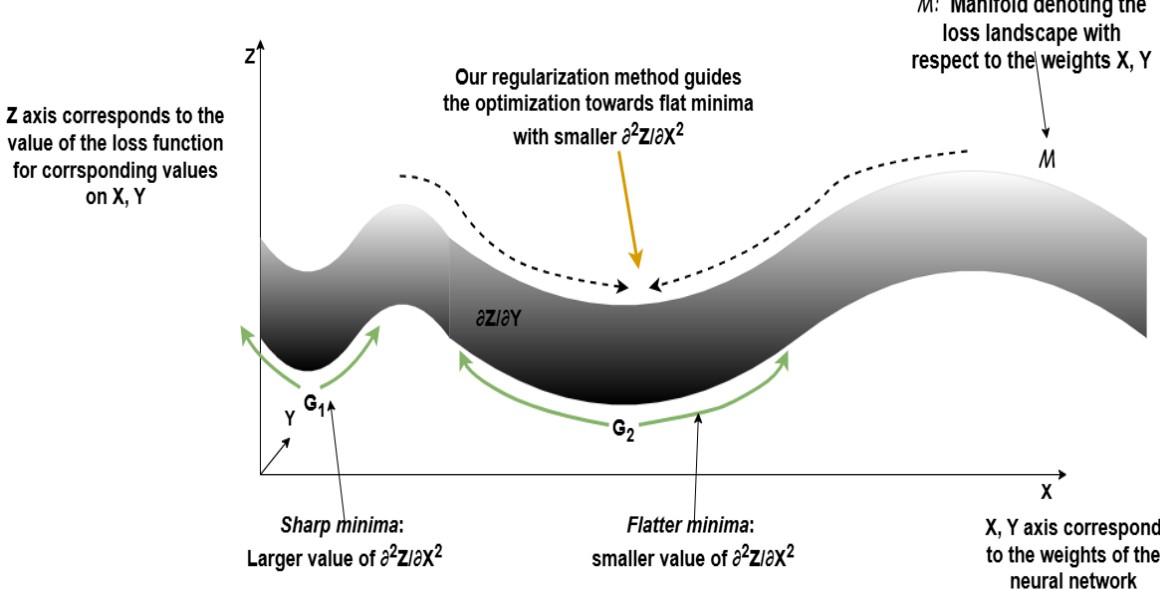

Figure 1: Figure demonstrating the properties of the solution learnt via our Fisher regularization. $X$ and $Y$ axes correspond to two weights of a neural network, whereas $Z$ corresponds to the loss for corresponding values of $X$ and $Y$. $G_1$ and $G_2$ corresponds to two different global minima where $G_1$ is a sharp minima, while $G_2$ is a flatter minima. Our regularization method will favor the minima near the flatter region of the landscape, due to its lower value of the Fisher term $\partial^2 z/\partial x^2$ corresponding to it.

with 10000 images . The CIFAR-100 dataset also contains 60000 images, but belongs to 100 categories, with 50000 images in the training set and 10000 in the test set. There are only 500 images per category in the training set, making it a more challenging dataset.

## 5.2 Data Preprocessing

For all the datasets, as a pre-processing step, we first normalized the images by subtracting the mean from each image pixel in each channel and dividing them by their variance. The mean and variance were calculated across all pixels in the training set across channels, and the images in the test set were normalized using the same statistics. The images in the CIFAR-10 dataset were augmented by vertical and horizontal shifts of 10% of the image size, and random rotations by a maximum angle of 15°, while no augmentation was applied to the MNIST dataset. The images in the CIFAR-100 dataset were pre-processed by first padding them with 4 pixels on either side and taking a $32 \times 32$ random crop on which a random horizontal flip was applied.

## 5.3 Network and experimental configurations

To evaluate the proposed method, we performed experiments on these datasets with deep convolutional SVM models with different architectures (Simonyan & Zisserman, 2014). On the MNIST dataset, we experimented with a network consisting of 3 convolution layers, each with a kernel size of 3, with successive layers containing 68, 128, and 384 hidden units, respectively. The outputs of the convolutional layers were batch normalized, and an $L_2$ regularization of 0.0003 was applied to their weights. Each convolutional layer was followed by a max-pooling layer of size $2 \times 2$. The output of the final convolutional layer was flattened and passed to a fully connected layer with 800 hidden units and ReLU activations. The output from this layer was passed to a linear layer with $c = 10$ units ($c$ = number of classes) that served as the output of the SVM classifier. Since our model was aimed at solving the multiclass classification, we used the multiclass variant of the hinge loss (Moore & DeNero, 2011) (or the margin loss) with margin $\gamma = 1$, which is defined as follows:

$$\sum_{i=1}^{n} max(0, 1 + w_{y_i} x_i - \max_{t \neq y_i} w_t x_i) \tag{8}$$

where $y_i$ denotes the actual class labels and $\max_{t \neq y_i} w_t x$ denotes the maximum output among the classes other than the ground-truth class.

For the CIFAR-10 dataset, we followed the AlexNet model described in (Krizhevsky et al., 2012a) after reducing the kernel size of its first convolution layer from 11 to 5, as well as the VGG-19 model (Simonyan & Zisserman, 2014). We used the Leaky ReLU (Maas et al., 2013) with a negative slope of 0.5 as our activation function instead of the ReLU activations, as the ReLU caused the problem of vanishing gradient. We batch-normalized the output at every layer. No dropout or regularization was applied at any layer. For the CIFAR-100 dataset, we used a DenseNet-40 model (Huang et al., 2018) with a growth rate of 12. A dropout of 0.4 was applied to all layers along with a weight decay of $1e - 4$ both of which were crucial to obtain decent results.

The networks were trained using backpropagation through stochastic gradient descent. A sum of the multiclass hinge loss (Tang, 2013) with the regularization term $r(w)$ was used to train the SVM classifier. The overall loss function for the classifier can be written as

$$\sum_{x_i, y_i} max(0, 1 + w_{y_i} x_i - \max_{t \neq y_i} w_t x_i) + r(w) \tag{9}$$

where $x_i$ and $y_i$ iterate over all pairs of data points and their corresponding labels, $r(w)$ denotes the regularization function, and $w$ denotes the weights.

We trained the models using four regularization schemes:

- The vanilla $L_2$ regularization, where the regularization term $r(w)$ in equation (9) reduces to

$$r(w) = \beta \|w\|^2 \tag{10}$$

- the *spectral* norm regularization, which penalizes the largest singular value of the weight matrices to make the network output less sensitive to small perturbations. Here, the regularization term is $r(w) = \gamma \cdot \sigma(W) + \beta \|W\|^2$, where $\sigma(W)$ denotes the spectral norm, or the largest singular value of the weights $W$.

- The $group - l_1$ (or the group lasso)(Scardapane et al., 2017), with the regularization penalty $r(w) = \gamma \cdot \sigma(W) + \beta \|W\|_{2,1}$, where $\|.\|_{2,1}$ denotes the $L_{2,1}$ norm and finally,

- our Fisher regularization, which had the regularization penalty:

$$r(w) = \gamma \mathbf{w} \mathbf{F} \mathbf{w}^{\mathbf{T}} + \beta \|w\|^2 \tag{11}$$

In all the regularization schemes with which we experimented, we always added a small amount of $l_2$ penalty $\beta \|W\|^2$ to allow the maximization of margin ($\frac{1}{\|\mathbf{w}\|^2}$) in the SVM objective. The $group - L_1$ regularization method penalizes the $L_{2,1}$ norm $\|W\|_{2,1} = \sum_{i=1}^{n} \sqrt{\sum_j W_{i,j}^2}$, which applies a $L_2$ norm to the columns, followed by a $L_1$ norm to the resulting output. Since the weights in a column correspond to those emanating from the same hidden unit, taking an $L_1$ norm over the column norm allows the network to learn weights that are sparse across columns, i.e., only a few columns have non-zero weights. This allows the network to *prune* less valuable hidden units, reducing network complexity while improving generalization.

The generalization gap was evaluated by tracking the difference between the training and test losses of the trained models. Furthermore, we normalized the terms in the Fisher matrix to scale their magnitude between 0 and 1 as follows

$$\|\frac{\partial L(X)}{\partial w_i} \cdot \frac{\partial L(X)}{\partial w_j}\| = \frac{\|\frac{\partial L(X)}{\partial w_i} \cdot \frac{\partial L(X)}{\partial w_j}\| - \min_{j,k} \|\frac{\partial L(X)}{\partial w_k} \cdot \frac{\partial L(X)}{\partial w_j}\|}{\max_{j,k} \|\frac{\partial L(X)}{\partial w_k} \cdot \frac{\partial L(X)}{\partial w_j}\| - \min_{j,k} \|\frac{\partial L(X)}{\partial w_k} \cdot \frac{\partial L(X)}{\partial w_j}\|} \tag{12}$$

This normalization rescales the terms of the Fisher matrix between 0 and 1 when they are either extremely large or small, and also allows us to consider the *relative* scale by which to penalize each term in the regularization described in (10). For all experiments, we recorded the generalization gap (or the difference between the training and test losses) of the model.

From the experiments, we observed that training with our Fisher regularization method improved the test accuracy of our network compared to the regular $L_2$ penalty. The values of $\beta$ and $\gamma$ for the different regularization methods are given in the following table.

| Model | Regularization hyperparameters ($\beta/\gamma$) | | | |
|---|---|---|---|---|
| | $l_2$ | *group* | *spectral* | *fisher* |
| 3 layer CNN SVM (MNIST) | 0.007/0 | 0.007/0.01 | 0.007/0.01 | 0.007/0.01 |
| AlexNet SVM (CIFAR-10) | 0.05/0 | 0.04/0.005 | 0.04/0.01 | 0.03/0.12 |
| VGG-19 SVM (CIFAR-10) | 0.05/0 | 0.04/0.005 | 0.05/0.01 | 0.05/1.0 |
| DenseNet-40 SVM (CIFAR-100) | 0.0001/0 | - | - | 0.0001/0.002 |

Table 1: The regularization parameters $\beta$ and $\gamma$ for each of the model and the corresponding regularization method selected after cross-validation. The parameters are mentioned as $\beta/\gamma$ corresponding to each regularization technique and the model.

These hyperparameters were selected using the cross-validation technique by monitoring the performance of the model on a holdout set for a fixed number of epochs.

We trained the model for 40 epochs on the MNIST dataset with a batch size of 200. On the CIFAR-10 dataset, we trained the AlexNet and VGG-19 models for 55 and 80 epochs with a batch size of 64, and on CIFAR-100, we trained a DenseNet-40 model for 114 epochs with a batch size of 64. Adam optimizer (Kingma & Ba, 2014) with a learning rate of 0.0001 was used to train all networks. The learning rate was decayed exponentially with the number of epochs according to the following formula.

$$lr(t) = lr(0) \cdot (0.5^{\lceil t/lr\_drop \rceil}) \tag{13}$$

where $t$ is the epoch number and $lr\_drop$ is the number of epochs for which the factor $\lceil t/lr\_drop \rceil$ stays constant, after which the learning rate drops by a factor of 0.5. We used an $lr\_drop$ of 10 for the MNIST dataset, $lr\_drop$ of 20 and 15 on the AlexNet and VGG-19, respectively, for the CIFAR-10 dataset, and 38 on the DenseNet-40 model for the CIFAR-100.

The performance and accuracy of all the models trained with different regularization methods are reported in Table 2 for the MNIST and in Table 3 for the CIFAR-10 dataset.

## 6 Results

The test accuracy and generalization performance of the different SVM models, trained with our Fisher regularization, are compared with that of simple $L_2$ regularization, spectral norm regularization, and the group $l_1$ regularization in Table 2 for the MNIST dataset and Table 3 for the CIFAR-10 and 100 datasets. For each model, the test accuracy and the generalization gap (difference between test loss and train loss)

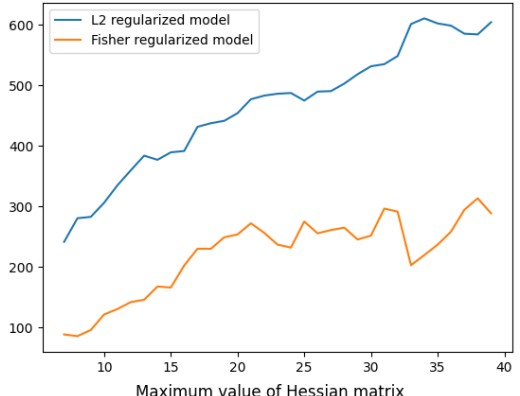

Figure 2: Figure illustrating the evolution of the maximum value of the Hessian across the training epochs on the MNIST dataset. Each point in both the line graphs shows a moving average of this value over the last 8 epochs to filter out the effect of noise. As can be seen from the graph, this value continues to increase rapidly for the L2 regularized model, whereas it increases and then becomes relatively stable for the Fisher regularized model.

on the test set are mentioned in the respective tables. On the MNIST dataset, we first observe from Table 2 that models trained with our Fisher regularization achieved a better test accuracy of 99.43% on the test set than the SVM model trained with standard $L_2$ regularization (99.28%), as well as the spectral norm and group-$l_1$ norm regularization methods. In Figure 2, we can also observe that our Fisher regularized model had a lower value of the maxima of the Hessian matrix compared to that of the model trained with $L_2$ regularization on the MNIST dataset.

Similarly, on the CIFAR-10 dataset, our Fisher regularized AlexNet model achieved 83.29% test accuracy, surpassing the accuracy of models trained with the $l_2$, spectral, and $group - l_1$ regularization methods by 0.82%, 0.65%, and 0.43%. Moreover, our Fisher regularized model outperformed all the other regularization methods for the VGG-19 architecture on the same dataset. On the more challenging CIFAR-100 dataset, a DenseNet trained with our regularization method achieved a 67.74% accuracy, which was 0.45% higher than that achieved with standard $L_2$-norm weight decay (67.29%).

These results demonstrate that the models trained with the Fisher regularization method achieved better generalization than the models trained with other regularization methods. Furthermore, the generalization gap was smaller for our models compared to the models trained with $l_2$ regularization across all models trained on the CIFAR-10.

| Model | Accuracy |
|---|---|
| Convolutional Network SVM (L2 regularized) | 99.28 |
| Convolutional Network SVM (Spectral norm) | 99.34 |
| Convolutional Network SVM (group L1 regularized) | 99.40 |
| Convolutional Network SVM (Fisher regularized) | **99.43** |

Table 2: Test accuracy comparison of the Convolutional network SVM model (Krizhevsky et al., 2012b) trained with our Fisher regularization, with the other regularization approaches on the MNIST dataset. (Lecun et al., 1998)

| Model | Test Accuracy | | | | Generalization Gap | | | |
|---|---|---|---|---|---|---|---|---|
| | $l_2$ | *group* | *spectral* | *fisher* | $l_2$ | *group* | *spectral* | *fisher* |
| AlexNet SVM | 82.47 | 82.86 | 82.64 | **83.29** | 0.2595 | 0.1815 | **0.1023** | 0.1409 |
| VGG-19 SVM | 88.29 | 87.86 | 87.87 | **88.55** | 0.2770 | 0.0513 | 0.0721 | **0.1484** |

Table 3: Test accuracy comparison of the Convolutional network SVM models (Krizhevsky et al., 2012b) trained with different regularization approaches on the CIFAR-10 dataset. (Krizhevsky, 2009)

| Model | Accuracy |
|---|---|
| DenseNet-40 SVM ($L_2$ regularized) | 67.29% |
| DenseNet-40 SVM (Fisher regularized) | **67.74**% |

Table 4: Test accuracy comparison of the DenseNet SVM models (Huang et al., 2018) trained with our fisher regularization with that of $L_2$ regularized model on the CIFAR-100 dataset. (Krizhevsky, 2009)

## 7    Discussion

In this work, we proposed Fisher regularization, a novel method for improving the generalization performance of a deep neural network based on the Fisher norm instead of the commonly used $l_2$ norm regularization. The Fisher norm leverages the curvature or *sharpness* of the loss landscape into the regularization term and gives greater importance to weights aligned with high-curvature directions. We provided a theoretical justification for our regularization grounded in the PAC-Bayesian principles, which served as a basis for its observed empirical benefits. Extensive experiments were performed on several popular deep CNN architectures using different regularization methods, including our Fisher regularization. The result of our experiment demonstrated the superior performance and efficacy of our Fisher regularization scheme over standard $l_2$ regularization, spectral norm regularization, and $group - l_1$ norm regularization across standard image classification datasets. For example, when applied to an AlexNet-SVM model trained on the CIFAR-10 dataset, our regularization method achieved 0.82% and 0.65% higher test set accuracy compared to that achieved via $l_2$ and spectral regularization. Similarly, models trained with our approach achieved higher test accuracy on the CIFAR-100 dataset as well as the MNIST dataset compared to that achieved by other regularization schemes. Furthermore, the generalization gap of the models trained with our methods was smaller than that of $l_2$ weight decay across all architectures on the CIFAR-10 dataset. These findings support the fact that our Fisher regularization method improves the generalization performance of a model over other strong regularization techniques, including $l_2$, spectral norm, and group-$l_1$ regularization, as demonstrated by the higher test accuracy compared to other state-of-the-art regularization methods.

## 8    Conclusions and Future works

This paper introduced a novel and theoretically grounded regularization method, Fisher Regularization, which leverages the Fisher Information matrix to dynamically penalize the weights of a neural network based on the local geometry of the loss landscape. Unlike traditional regularization methods that give equal importance to all parameters of the network, our Fisher regularization penalizes the weights associated with higher curvature directions in the loss landscape more aggressively, thus guiding the optimization toward a more *flatter* and robust global minima.

We provide a rigorous theoretical justification for Fisher regularization using the PAC-Bayesian framework, demonstrating a tighter generalization bound based on our Fisher norm. This theoretical underpinning provides a strong rationale for the effectiveness of our method. Extensive experiments on diverse benchmark datasets demonstrated that deep convolutional SVM models trained with our method achieved consistently better performance than other methods on the test set of all the datasets, thereby empirically validating

the superiority of our approach. These observations demonstrate that our Fisher regularization can be a powerful tool that can be used to train neural networks that are not only highly accurate but also demonstrate improved generalization.

In the future, we would like to extend our approach to deeper models with better performance and apply it to other layers beyond the SVM classifier weights, such as those of the convolutional layer. Furthermore, we would explore its application in modifying the optimization algorithm (SGD) by guiding it to flatter and more robust minima. Finally, we would test it over other tasks such as fine-grained classification (Zhao et al., 2017) and semantic segmentation, and on other domains such as natural language processing.

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

## A Appendix

### A.1 Code to evaluate Fisher regularization term

The algorithm to compute the Fisher norm term is integrated into the *train_step* function of our code, shown below. This function belongs to our main keras.Model class, and is automatically invoked during training when the .fit() function is called on the model. The code snippet shown below describes the *train_step* function in detail, illustrating how it calculates the multiclass hinge loss and sum of our Fisher norm and $L_2$ norm, and then optimizes the model parameters by minimizing these losses.

```
@tf.function
def train_step(self, data):
    X, label = data
    with tf.GradientTape(persistent=True) as tape:
        pred = self.classifier(X); #Returns output of classifier
        cls_loss = self.cls_loss(label,pred) #Calculates the hinge loss
        grads = tape.gradient(cls_loss, self.classifier.trainable_variables[-2])
        #Calculates the gradient of the classifier layer with respect to the loss
```

```
grads = tf.reshape(grads,[-1,1])
#reshapes the gradient to a vector
grads_mul = tf.abs(tf.matmul(grads,grads,transpose_b=True));
#Computes the Fisher information matrix F (here, F = grads_mul)

ming = tf.reduce_min(grads_mul); maxg = tf.reduce_max(grads_mul);
grads_mul = (grads_mul-ming)/(maxg-ming+1e-8)
#Scales the terms of the matrix

w = self.classifier.get_layer(name='margin_dense').kernel
#Extracts weights of the classifier

w = tf.reshape(w,[8000,1])
fisher_norm = 0.01*tf.reduce_sum(tf.matmul(tf.matmul(w,tf.stop_gradient(grads_mul),
transpose_a=True),w)) #Computes fisher norm ‖w ⊗ F ⊗ w‖
l2_norm = 0.007*tf.math.reduce_sum(tf.reduce_sum(w**2)) #The regular L₂ norm
norm = l2_norm + fisher_norm #Adds the norms
cls_loss = cls_loss + norm #Adds the norm to the final loss term

grads = tape.gradient([cls_loss],self.classifier.trainable_variables)
# computes the gradient of each parameter with respect to the loss
self.optimizer.apply_gradients(zip(grads, self.classifier.trainable_variables))
# Optimizes the model parameters by applying the gradients
return {'cls_loss':cls_loss,"acc": accuracy,'margin':margin,'fisher_norm':fisher_norm}
```

### A.2    Adjusting hyperparemeters of the model during experiments

The hyperparameters of the model can be easily adjusted in the code demonstrated above. The strength of the different regularization terms (the $\beta$ and $\gamma$ parameters) can be adjusted by changing the constant multiplied with the term. For example, in the code shown above, a constant of 0.007 is multiplied to the result of the *l2_norm* term and can be adjusted to change the $\beta$ parameter and adjust the regularization strength. Similarly, the 0.01 constant multiplied with the *fisher_norm* term can be changed to adjust the hyperparameters as desired. The number of epochs for both training the model as well as the one used in the learning rate scheduler can be modified by changing the epochs in the model.fit() function and changing the number of epochs in the variable *lr_drop*, respectively.

