# OpenReview forum: "Enhancing generalizability of deep networks via Fisher regularization"
_TMLR — Rejected by TMLR_

### Review · Reviewer_vb2S · 2025-06-10

**Summary Of Contributions:**

The authors strive to introduce a regularization, Fisher Regularization, which penalizes the term $wFw^T$ in order to guide optimization toward flatter and more generalizable minima. Empirical validation on Cifar benchmarks demonstrates that models trained with Fisher Regularization outperform convergence methods.

**Audience:**

No

**Broader Impact Concerns:**

I have not found any discussions about the limitations and potential negative societal impact. But in my opinion, this may not be a problem, since the work only focuses on the optimization in deep learning. Still, it is highly encouraged to add corresponding discussions.

**Claims And Evidence:**

No

**Requested Changes:**

See weakness.

**Strengths And Weaknesses:**

**Strengths**

1. The paper is clearly written and easy to follow.
2. How to converge to flat minima is a significant area of interest in optimization.


**Weakness**
1. The benefit of flat minima for generalization has been a known fact for a long time. The authors do not provide any new insights beyond current well-established understanding.

2. The authors have not involved any work focused on training techniques that promote convergence to flat minima. Notably, Sharpness-Aware Minimization (SAM) is the most typical and well-known method, which has many variants. Additionally, gradient norm penalty have been shown to achieve similar effects. It is therefore not acceptable to omit a comparison with these closely related methods. The authors can easily find these works, so I will not list these here.

3. The authors are encouraged to review the experimental sections of the related literature. At present, the experimental design and validation can not meet the current standards in the field.

---

### Review · Reviewer_uh5R · 2025-06-13

**Summary Of Contributions:**

This paper aims to improve the generalization ability of deep learning models by regularizing on an extra Fisher information term. The authors leverage a Fisher information approximation for KL divergence between nearby distributions to acquire a generalization bound. The method is then tested on some widely adopted computer vision networks and datasets, showing some performance advantage compared to other traditional regularizations.

**Audience:**

Yes

**Broader Impact Concerns:**

No concerns.

**Claims And Evidence:**

No

**Requested Changes:**

* Clarify eq. (4) is an approximation instead of an equality.
* Use traditional network architectures and report the standard deviations to strengthen the experimental results.

**Strengths And Weaknesses:**

Strengths:
* The writing of the paper is clear and easy to follow. The design of the proposed method is well-motivated by the Fisher information approximation.

Weakness:
* The derivation of the generalization bound is not rigorous. Eq. (4) is actually an approximation based on Taylor series expansion, instead of an equality. Therefore, the bound in eq. (5) does not generally hold. This result should be regarded rather as an approximation of the original bound in eq. (2) than a generalization bound.
* The experimental results are not strong enough. The performance improvement is minimal compared to baseline methods. No standard deviation or confidence interval is reported for the accuracy scores. The usage of deep networks is also not standard. Why use SVM as the final classifier instead of the original full-connection layers? These facts affect the significance of the experimental results.

---

### Review · Reviewer_77Eo · 2025-06-29

**Summary Of Contributions:**

The paper introduces Fisher Regularization (FR) which explicitly penalizes the curvature of the loss landscape with respect to the final (output) layer of deep networks by adding a FR term $w F w^\top$ where $F$ is the Fisher information matrix and $w$ is the last layer weights. They provide a PAC-Bayesian bound on the generalization gap and show empirical results on realistic datasets and deep convolutional networks. Experiments are compared to other regularization techniques and demonstrate accuracy gains.

**Audience:**

Yes

**Broader Impact Concerns:**

See above.

**Claims And Evidence:**

No

**Requested Changes:**

See above.

**Strengths And Weaknesses:**

### Strengths

1. Previous PAC-Bayesian bounds on the loss are related to Fisher information, providing a theory-motivated learning algorithm.

2. Paper provides a detailed account of all experimental procedures.

3. Their method is compared against other common penalties across several architectures and datasets.

### Weaknesses and Suggestions

1. Fisher regularization is not a novel method (e.g https://arxiv.org/pdf/1711.01530, https://arxiv.org/pdf/2012.14193). In fact, previous methods are more general in that they impose flatness on the entire loss landscape with respect to all parameters instead of only final layers. Authors also fail to cite these works.

2. It is not clear exactly how the bound in Eq. (5) is tighter than previous bounds cited by the authors (https://arxiv.org/pdf/1707.09564 and https://arxiv.org/pdf/1703.11008). Authors should show exactly which bound from which references is being improved and provide a rigorous theorem and its proof.

3. Deep SVM sounds confusing. The paper trains models end-to-end while SVM assumes a fixed feature map.

4. Related to the first point, the paper should provide a detailed comparison to previous sharpness miminizing works such as natural gradients or SAM (https://arxiv.org/pdf/2010.01412, https://arxiv.org/pdf/2206.04920) which is the state-of-the-art to my knowledge.

5. In addition, some works claim that flatness is unnecessary for generalization (https://arxiv.org/pdf/1703.04933). What are the authors' comments on this?

6. The regularization term presented in the paper differs from its implementation where FI matrix is preprocessed by 1) taking the abs of its elements and 2) normalizing its elements between [0-1]. How do the results look like without this preprocessing? Also how do the authors choose the hyperparameters $\beta$ and $\gamma$ and how does the final result depend on them?

---

### Decision · Action_Editor_X7ZF · 2025-08-01

**Recommendation:** Reject

**Audience:**

Yes

**Audience Explanation:**

This paper could be of interest to researchers working on the theory of deep learning.

**Claims And Evidence:**

No

**Claims Explanation:**

The reviewers raised several concerns, including (1) the theoretical analysis relies on an approximation that may not hold under general conditions; (2) the proposed method lacks novelty, and the authors did not cite or compare against many relevant baseline methods; and (3) the experimental section is not sufficiently thorough to support the current claims. The authors have not provided a rebuttal or revisions to address these comments.